# Current Status of Quantum Chemical Studies of Cyclodextrin Host–Guest Complexes

**DOI:** 10.3390/molecules27123874

**Published:** 2022-06-16

**Authors:** Anna Helena Mazurek, Łukasz Szeleszczuk

**Affiliations:** 1Department of Physical Chemistry, Chair of Physical Pharmacy and Bioanalysis, Faculty of Pharmacy, Doctoral School, Medical University of Warsaw, Banacha 1 Str., 02-093 Warsaw, Poland; anna.mazurek@wum.edu.pl; 2Department of Physical Chemistry, Chair of Physical Pharmacy and Bioanalysis, Faculty of Pharmacy, Medical University of Warsaw, Banacha 1 Str., 02-093 Warsaw, Poland

**Keywords:** cyclodextrin, host–guest complexes, DFT, QC, quantum chemistry, density functional theory, CD complexes

## Abstract

This article aims to review the application of various quantum chemical methods (semi-empirical, density functional theory (DFT), second order Møller–Plesset perturbation theory (MP2)) in the studies of cyclodextrin host–guest complexes. The details of applied approaches such as functionals, basis sets, dispersion corrections or solvent treatment methods are analyzed, pointing to the best possible options for such theoretical studies. Apart from reviewing the ways that the computations are usually performed, the reasons for such studies are presented and discussed. The successful applications of theoretical calculations are not limited to the determination of stable conformations but also include the prediction of thermodynamic properties as well as UV–Vis, IR, and NMR spectra. It has been shown that quantum chemical calculations, when applied to the studies of CD complexes, can provide results unobtainable by any other methods, both experimental and computational.

## 1. Introduction

Due to their unique structural, physical, and chemical properties, cyclodextrins (CDs) and their derivatives have been of great interest for more than a century [1]. The biodegradability, biocompatibility, and versatility of CDs and CDs-based materials extend their applications to new areas every year; however, the main property that makes CDs so popular is their ability to form host–guest complexes with a variety of compounds [2].

CDs are commonly used in pharmaceutical formulations as they increase the solubility of poorly soluble drugs and protect substances against external factors, such as light, humidity, and heat [3]. CDs can mask unpleasant smells or flavors of drugs, which is especially important in formulations dedicated to children [4]. More than 100 original drugs are currently being manufactured with CDs as excipients [5,6,7].

Interactions between CDs (host) and guest molecules may yield a stable complex with a high equilibrium constant; for example, β-CD forms highly stable inclusion complexes with adamantyl derivatives with a binding constant of ~10^4^–10^5^ M^−1^ [8,9]. It is not surprising then that the number of newly obtained cyclodextrin host–guest complexes is constantly increasing. However, only a small amount of those complexes is being reported with their crystal structures. This is caused by the fact that most of those complexes are either amorphous or polycrystalline, and even for the crystalline complexes it is usually very hard to obtain a crystal of a size suitable for single crystal X-ray measurements [10,11]. This is probably one of the reasons why a lot of experimental works describing the structure and properties of CDs complexes are supported by theoretical calculations.

To fully understand the behavior and physicochemical properties of a complex, knowledge of its structure is essential. However, it is not just the desire to reveal how the complex looks that makes the application of molecular modeling methods so popular in the studies of CDs complexes. By choosing an appropriate computational approach, it is possible to determine (or explain) the molar stoichiometry of the complex, the differences observed in the spectra (UV–VIS, IR, NMR) of host–guest physical mixtures and their complexes, and also to predict the stability of such structures under various conditions such as different solvents, temperature or pressure.

CDs host–guest complexes are surely very flexible structures, which is the common cause of their polycrystallinity. This is why a lot of the molecular modelling studies devoted to those complexes utilize molecular dynamics simulation at the molecular mechanics level. Those works have been recently reviewed by us [12]. However, the types of intermolecular forces that stabilize such complexes such as hydrogen bonding, van der Waals, and hydrophobic and dipole–dipole interactions, usually cannot be modeled with the required accuracy using the molecular mechanics methods. This is why the number of works in which the calculations of CD complexes at the higher level of theory, namely, quantum chemical (QC), has constantly increased since 2005 (Figure 1). Now, after 20 years of studies, the number of such articles is large enough to draw some general conclusions and trends as well as the advantages and disadvantages of such an approach. Therefore, the aim of this review was to gather and analyze the works in which the CD host–guest complexes have been modeled using QC methods.

There were at least a few reasons behind writing this review. First, it was worth analyzing whether some conclusions can be made on the choice of the most accurate method, including the applied DFT functional, basis set, dispersion correction or solvent model. Those aspects are discussed at the very beginning. Then, it was interesting to check the main reasons behind the QC method chosen by the authors of the reviewed works. Was it solely to predict or suggest the possible structure of the complex, or were the calculations used for something more such as the simulation of the spectra or explanation of the reaction mechanism to support the experimental findings? Further, we wanted to find out what type of CDs and what possible guests were studied in those computational works. Therefore, an informative table presenting the most essential information such as the composition of the complex, applied functional, basis set, and solvent treatment has been prepared to serve as an informative and easy-to-follow guide for future studies. Finally, the chosen examples are described in a more detailed way, suggesting the possible solutions and future indications.

As the authors of this review have been using the QC methods to model the structure and properties of CD complexes and found this approach very useful, it was our hope and desire to convince other researchers to try such solutions in their works. 

## 2. Applied Calculational Methods and Parameters

### 2.1. Choice of QC Method

The computational methods that were used in the reviewed works nicely correspond with the general increase in the computational power available to researchers worldwide. The earliest (before 2005) QC works studying CD complexes were done using the least demanding semi-empirical methods such as AM1, PM3 and later PM6 or PM7. Subsequently, those methods have been gradually superseded by density functional theory (DFT) calculations, while recently a few works have been published in which the Møller–Plesset perturbation theory (MP) was applied. It is worth noticing that CD complexes are not small objects, in terms of QC calculations, especially when the γ-CD or substituted CDs are the hosts with the large ligand as a guest. Application of QC can also be a problem when studying complexes with a host:guest ratio higher than 1:1. This is why even in the 2020s in some works, semiempirical methods have been applied; however, the ratio of DFT to the semi-empirical ones is constantly increasing (Figure 2). 

### 2.2. General Remarks

Calculations of CD other than in the form of typical complexes

As stated in the title of this manuscript, this review focuses on the application of QC methods in the analysis of CD complexes. However, computational studies are also performed for the systems where CDs play different roles, i.e., as nanocarriers, corrosion inhibitors or building blocks for even more complex structures, often with some additives such as gold particles. Those studies have not been listed in Table 1. The selected newest articles on the topic are [13,14,15,16,17,18]. 

It should be noticed that to increase the accuracy of DFT calculations in solid state, periodic boundary conditions are often applied, using crystal unit cells as simulation boxes. During the computations, only the properties of the original unit cell need to be calculated and then propagated in the chosen dimensions. However, to perform such computations, the crystal structure of the studied object is mandatory. More information on such calculations can be found in a recent review [19] with an example of such calculations for CDs presented in [20].

Software

In almost all of the reviewed works presenting the results of the DFT calculations, Gaussian software was used. There are only a few cases when ORCA [21,22,23], VASP [24] or ADF [25] were applied instead. Only for the solid state calculations or those involving adhesion, nanocarriers, etc., is DMol3 also commonly used. 

ONIOM

In several works that included CD where DFT was applied, the ONIOM method was presented (see Table 1). ONIOM stands for Our own N-layered Integrated molecular Orbital and Molecular mechanics and is a hybrid method which combines QC (either ab inito or semi-empirical) and molecular mechanics methods in order to reduce the computational cost [26]. On the basis of the ONIOM results, other properties such as thermodynamic ones are calculated. This approach was for years popular in the computation of the CD systems, as CDs are relatively big structures, and for a long time it was not possible to calculate both the CD and a guest using DFT. Therefore, CD was considered an outer layer and there a lower level of theory was applied, and the guest molecule was computed using a higher level of theory. However, with the general increase in computational power available, this approach is currently rarely used in the studies of CD complexes due to its lower accuracy when compared with pure QC calculations.

Molecular dynamics

An MD simulation is a well-established technique used for the study of various molecules complexes and mixtures in any state of matter and at almost any temperature and pressure condition. It can be used to determine structural, energetic, and thermodynamic properties as well as a means to scan the potential energy surface of a studied system. 

For MD simulations of large molecular complexes, such as ligand–protein, molecular mechanics (MM) methods are commonly used. On the contrary, when MD simulations are performed on relatively small molecules, it is usually at the quantum mechanics (QM) level of theory, which significantly increases the accuracy of calculations, but also their computational costs. In terms of the sizes of the modeled objects, CD complexes are somewhere in between. While geometry optimization calculations on the static structures of CD complexes are, nowadays, performed mostly at the QM level, usually by the means of DFT, the MD simulations are still being performed at the MM level [12].

### 2.3. Semi-Empirical Methods

Semi-empirical methods are based on the Hartree–Fock equation but simplified by the application of the empirical corrections [27]. Semi-empirical calculations are much faster than their ab initio counterparts, mostly due to the use of the zero differential overlap approximation. Their results, however, can be very wrong if the molecule being computed is not similar enough to the molecules in the database used to parameterize the method. Here, we will concentrate on the most popular semi-empirical methods, that is AM1, PM3, PM6, and PM7. 

According to Figure 2, the most popular semi-empirical method applied for the cyclodextrin complexes is PM3. However, it may be argued whether PM3 delivers better results than the newer generations, PM6 and PM7. The reason for wider application of PM3 instead of PM6 and PM7 may be the fact that some researchers got used to PM3 and some software does not support the newer parametrizations (PM6 and PM7). Since 2019, the DFT methods have strongly overtaken the role of a leading quantum chemistry-based calculation method in the cyclodextrin systems. Naturally, DFT is more precise than any semi-empirical approach, which leaves no space for further investigation, and among the semi-empirical methods, seems to be the most reliable in the complexes in question. Below, the application of the semi-empirical methods in the cyclodextrin-including complexes is described based on the examples from 2015–2022. However, it should also be noticed that the PM3, PM6, and PM7 approaches often co-exist with the DFT ones. In those cases, the systems are first optimized with the semiempirical method to obtain the initial structure for the DFT calculations. 

The oldest of all here presented semi-empirical methods is the AM1 approach. There are just a few examples of the cyclodextrin complexes calculated using this approach. In some of them, AM1 was applied to perform the geometrical optimization of a whole guest–CD complex [28]; in others, both substrates were calculated by AM1 whereas the complex underwent the DFT treatment [29], and in others, AM1 was applied solely for the CD. 

The next generation of semi-empirical methods is PM3. There are studies showing that PM3 predicts the presence and energy of hydrogen bonds better than AM1 [30]. Moreover, it has been reported that after α- and β-CD optimization with AM1 and PM3, the former resulted in badly distorted geometries, whereas the latter reproduced the crystalline structure rather well [31]. 

Semi-empirical methods are repetitively reported to deliver good insight into the complex formation process as well as reliable order of the configuration stabilities [32,33,34,35]. These methods help to determine global or local minima. However, it is stressed that to obtain reliable complexation energy values, DFT calculations should be performed [36]. Often, the PM3 approach is undertaken along with the ONIOM DFT/PM3 approach [34,37,38,39,40,41,42,43,44]. 

The most often and standard PM3 application in the cyclodextrin complexes is to move the guest along the selected axis going through the CD cavity. The guest is stopped every 1 Å, usually between −8 Å or −10 Å and respectively +8 Å or +10 Å. Additionally, the guest is rotated from 0°to 360° usually every 20° or 45°. At each such stopping point, the complexation energy is measured with PM3 [37,44,45,46,47].

In the way of their application to the cyclodextrin systems, the PM6 and PM7 approaches follow the same pattern as PM3. Namely, they are used to gain insight into the complex structure [48] and thermodynamic properties [48,49,50,51,52], which allows to determine the most stable complex [52,53,54,55], and, as in case of ofloxacin enantiomers, rank the eluted substances in the order in which they will be eluted [49]. Similarly, as in the case of PM3, FT-IR spectra can be simulated [50,56]. Further, PM6 and PM7 are often combined with the DFT methods in form of the ONIOM approach [42,52,57,58]. 

In contrast to PM6, in PM7, the description of dispersion interactions and hydrogen bonding has been improved, and consequently the errors associated with modelling large molecules and complexes have been reduced [49]. The description of properties such as heat of formation or height of the reaction barrier has been improved [59]. In turn, it has been reported that PM6, compared with PM3, can yield better agreement with the experimental values [59]. In another study, when compared to the experiment, PM3 provided wrong and at the same time opposite results to those obtained by PM7 [60]. To depict another example, in a study where β-CD, dimethyl-β-CD, and hydroxypropyl-β-CD were analyzed by both PM6 and PM7, in all cases PM7 delivered complexation energies of significantly lower values [58]. 

A separate topic is ADMP, the Atom Centered Density Matrix Propagation Molecular Dynamics approach, which can be performed with semi-empirical, Hartree–Fock or DFT methods. It provides equivalent functionality to Born–Oppenheimer molecular dynamics at a considerably reduced computational cost. The ADMP method has a number of attractive features. Systems can be simulated by accurately treating all electrons or by using pseudopotentials. Through the use of a tensorial fictitious mass and smaller values of the mass, reasonably large time steps can be employed, and lighter atoms such as hydrogens need not be replaced with heavier isotopes. A wide variety of exchange-correlation functionals can be utilized, including hybrid density functionals [61]. In the last decade, only two cases have been published: β-CD-olsalazine with PM3-ADMP [62] and β-CD-propranolol with PM6-ADMP, ONIOM(DFT/PM3)-ADMP, and DFT-ADMP approaches [63]. ADMP results confirm the importance of the non-bonded interactions in the complex stabilization.

To sum up this part of the review, a relatively new and complex study should be cited. In the article entitled ‘Prediction of correct intermolecular interactions in host–guest systems involving cyclodextrins’ [63] published in 2020, the following approaches were tested: AM1, PM3, PM6, and DFT with the most standard B3LYP/6-31G(d,p), as well as PM6 and DFT with and without dispersion correction. The study involved 15 α-CD and 28 β-CD inclusion complexes in terms of both geometrical parameters and complexation energy values. The results showed that the most accurate was the B3LYP/6-31G(d,p)-D3 approach, followed by PM6-D3. Nevertheless, taking into account the high computational requirements of the DFT methods, the authors suggest that PM6-D3 is the most accurate and cost-effective approach. However, it must be stated that as the availability of the computational power is developing quickly, the DFT approach might shortly be, or already is, the best option for the computational analysis of the structure and energy of CD systems.

### 2.4. Density Functional Tight Binding (DFTB)

An approach that can be positioned between semi-empirical methods and DFT is the Density Functional Tight Binding Self Consistent Charge method (DFTB-SCC, referred here to as DFTB), which is often described as DFT approximation [64]. According to the Web of Science, just a few articles referring to DFTB and CD complexes have been published. However, those works show a relatively wide spectrum of possible DFTB applications. In the oldest works [64,65], DFTB was applied to verify the experimental NMR results and deliver some additional structural information. In the first case [64], the method’s application confirmed which conformation of spironolactone was preferred in the complex. In the second case [65], DFTB confirmed that in the analyzed peptide, tyrosine was a favored residue to access the CD’s cavity. In this second study, DFTB was applied within the QM/MM approach. In both of the described cases, the authors claimed good agreement of the obtained computation data with the experimental data. 

A more complex situation is described in [66], where the topic is self-inclusion (in the own cavity) of the CD’s substituents. Here, DFTB was compared with the DFT approach. DFTB is said to be method of choice as it predicts the stability order of the analyzed complexes properly, delivers the energy data that are close to the DFT data, and is faster than DFT. Both options including dispersion correction and the absence of this correction have been tested in DFTB and in DFT. The results plainly show that in both cases, application of the correction is necessary. In the case of DFTB, the dispersion-corrected version of calculations result in lower complexation energies and the order is maintained. Nevertheless, the authors emphasize that the application of an empirical dispersion correction may significantly overestimate dispersion interactions, and therefore a comparison with a rigorous DFT method should be done. 

Some drawbacks of DFTB have been pointed out in the work that targeted the largest CD for which the crystallographic structure is known [67]. The heavy atoms’ RMSD between the optimized structure and the crystallographic one were between 0.89 Å and 1.35 Å for DFT, depending on the parameters applied, with the best results for the B3LYP functional and 0.95 Å for DFTB. However, even if the overall RMSD looks good, large discrepancies in the angles sizes when compared with the experiment have been reported in the case of the DFTB approach as opposed to all DFT methods. 

In turn, the article from 2018 utilizes the DFTB approach to analyze the tautomerization process during encapsulation of genistein in CD [68]. The results are clear and deliver important information on the complexation, namely, ‘DFTB-based MD simulations reveal that spontaneous keto-enol tautomerization occurs even within a hundred picoseconds, which suggests that the encapsulated genistein is complexed in the ordinary enol form of the drug molecule’.

### 2.5. Density Functional Theory (DFT)

Functionals

While performing the DFT calculations, the main two parameters that must be decided on are type of functional and a basis set. Functionals mathematically define the electronic energy, which when added to the kinetic and electrostatic energy of the system, sums up to the total system’s energy [68,69]. According to the literature (see Table 1) for the systems that included CD, the hybrid B3LYP [70], semi-empirical GGA (generalized gradient approximation): wB97XD, B97D3 [71] or meta-GGA kinetic energy density incorporating Minnesota [72] functionals have been applied so far. Among the last category, M06-2X, M05-2X, and M06-L are used, with M06-2X being the most common in the analysis of the non-covalent interactions, whereas M05-2X includes 0% Hartree–Fock (HF) exchange, and M06-2X has 54% HF exchange [73]. In one of the studies including CDs, it was concluded that M06-L delivered poor results [74]. This was expected as the analyzed complex was β-CD-alprazolam, whereas M06L has been designed for calculations of the systems including transition metals, inorganic or organometallics [21]. wB97X and 97D3 are comprised of 22% Hartree–Fock exchange in the short range and 100% Hartree–Fock in the long range [75].

Dispersion correction

Noncovalent forces, such as hydrogen bonding and van der Waals interactions, are crucial for the formation, stability, and function of most CD complexes. At present, ubiquitous van der Waals interactions can only be accounted for properly by high-level quantum-chemical wavefunctions or by the Quantum Monte Carlo method. In contrast, the correct long-range interaction tail is absent from all popular local-density or gradient corrected exchange-correlation functionals of DFT, as well as from the Hartree–Fock (HF) approximation. A long-range electron correlation effect, known as the London part of the dispersion energy term, is not included in the Kohn–Sham DFT equation [76]. For years this was an issue affecting the accuracy of the DFT calculations. Nowadays, several dispersion correction methods are available. Nevertheless, their inclusion not always improves the calculation effect, hence this should be tested separately for each system in question. The most widely used dispersion corrections are TS (Tkatchenko-Scheffler) [77], GD (Grimme Dispersion, written also as D) [78], and MBD (Many-Body Dispersion) [79]. However, to perform calculations on the systems that included CD, almost solely the semi-empirical Grimme dispersion correction was applied (see the Table 1). It occurs in the D2, D3, D4, and D3(BJ) versions, where BJ indicates Becke Johnson damping. This last one is rarely used in CD-complex calculations. It is claimed that ‘the damping function in DFT-D methods has only a minor impact on the quality of the results’ [80] and even the comparison between D3 and D3 (BJ) published by Stefan Grimme clearly indicates that ‘the differences between the two methods are much smaller than the overall dispersion effect’ [80]. D3 includes less empirical input than D2, can be called a newer D2-version, and is the dispersion correction that is currently the most widely used. 

No dispersion correction is applied to the Minnesota functionals that are parametrized for dispersion. The same applies to wB97X (written also as ωB97X), which is the dispersion-incorporating version of B97X. B3LYP is sometimes used as B3LYP-CAM (Cambridge extension) [81], which includes the long-range correction; however, this is not common, and there is no strong evidence that this type of dispersion correction gives better results than application of the Grimme correction. 

Sometimes it seems reasonable to perform the same calculations with two chosen functionals as has been done for the β-CD-2,2′-bipyridine complex [82]. The authors’ conclusion is that wB97XD showed reliability in elucidating weak interactions, whereas B3LYP allowed one to achieve a good time–precision compromise. 

Another interesting example is comparison of three types of functionals performed for the β-CD-procaine HCl system [83]. According to the authors, B3LYP showed the highest efficiency and quality of results, wB97XD was specifically used to analyze the long-range interactions, and M06-2X was applied to predict the presence of hydrogen bonds. On the other hand, there are works such as [84] (β-CD-8-Anilinonaphthalene-1-sulfonate) where it is claimed that among tested functionals, B3LYP, wB97XD and M06-2X, the overall best results were delivered by wB97XD.

To take one more example, for the β-CD-benzocaine system [53], where B3LYP, CAM-B3LYP, M05-2X, and M06-2X were tested, M06-2X was claimed to deliver excellent results when used to obtain the NMR spectra. In another study, for UV–Vis spectrum simulation, the B3LYP-D3-based results showed the best agreement with the experimental data. The tested functionals were BLYP-D3, B3LYP-D3, and M06-2X-D3 [22]. In the case of β-CD-5-fluorouracil, inclusion of the dispersion correction changed the interaction energy by 15% and by 20% in water and ethanol solvent, respectively [85].

This only allows us to draw three important conclusions. Firstly, a couple of functionals, preferably representing all three groups (hybrid, GGA, Minnesota) should be tested for each system. Secondly, the choice of a functional depends on the goal of the study: geometry optimization, thermodynamic parameters, NMR spectra, etc. Thirdly, this literature review sends a clear message that the three most used and effective functionals are B3LYP-(D3), wB97XD, and M06-2X. 

Basis set

It should not be forgotten that in the study preparation, the functional and dispersion correction and the basis set choice play a significant role. The basis sets applied for the CD complexes so far are the Pople, correlation-consistent (cc-pVDZ), and Karlsruhe (def-TZVP, def2-SVP) basis sets (see Table 1). However, the Pople ones are definitely the most common and the variety among them is large, for instance, 6-31G(d), 6-31G(d,p), 6-31+G(d,p), 6-311++G(d,p), 6-31G**, where ‘1′ means basis set enlargement, ‘+’ means an additional diffuse function, and ‘*’ means a polarization function. The choice among different Pople basis sets depends partly on the available computational possibilities and partly on the type of studied objects. Inclusion of diffuse functions is needed to properly calculate the long-range interactions, such as hydrogen bonds. In turn, by extending the size of a basis set, the addition of polarization functions is meaningless. This is why, in the works presented in Table 1, the most presented basis sets are 6-31G(d,p) and 6-311G(d,p). 

In Table 1, it is noticeable that the ‘6-31+G* for H, N, O and 4-31G for C’ basis set combination is present. However, first of all, several of these works have been published with one affiliation, in other words, there is one laboratory that uses such an approach, and secondly, such a combination of basis sets for one system has been used mainly in the past when insufficient computation possibilities were at hand. In order to perform computation using the limited available tools, it is common practice to perform the geometry optimization in a lower basis set and later, for instance, for single point calculations, a larger basis set is used. An analogical approach applies to the more and less computationally demanding functionals. 

### 2.6. Solvent

The last parameter to decide on refers to the environment of the system. Calculations can be performed in gas or in solvent. To simulate a solvent in DFT, implicit solvent models are typically used. For the CD systems, the most popular is the family of the Polarizable Continuum Models (PCM) [86]. The other possibility is the Solvation Model Based on Density (SMD). IEFPCM [87] is a reformulation of the dielectric PCM, and the C-PCM is a conductor-like PCM, closer to the COSMO model. SMD defines the free energy of solvation via two components: the one is electrostatic contribution arising from the self-consistent reaction field, the other comes from the short-range interactions between the solute and solvent molecules [88].

Both PCM and SMD treat the solvent as a continuum because using Quantum Mechanics, it would not be possible to calculate a system with a large number of explicit solvent molecules. However, this issue can be partly approached as has been done in the study [64]. There, for general calculations, IEFPCM has been applied, but additionally a separate set of calculations has been done on the limited number of water molecules placed inside the CD cavity. Such an approach is useful if there is the probability that the CD–guest interaction is influenced significantly by the solvent’s presence. For example, when it is assumed that the hydrogen bonds between the guest and host molecules are water mediated. Sometimes, inclusion of the solvent effect decreases the complexation energy significantly, as in the case of the dexamethasone and SMD model [89]. However, it must be pointed out that inclusion of a solvent in the calculated system not always results in better (closer to the experimental data) complexation energies.

Taking into account all what has been written above, in order to make a fully justified selection of the parameters for the DFT calculations, a cross-study including different but most commonly used functionals, dispersion corrections (presence or absence), basis sets, and the environment (gas or solvent, type of solvation) should be performed. To the authors’ best knowledge, so far no such study has been undertaken on any system that includes CD. The already published benchmark studies are quite uncommon and usually focus on modification of one of the parameters, i.e., functionals, dispersion correction or the solvation method [67,90,91,92].

Only afterwards, with the parameters chosen carefully for the analyzed system (e.g., CDs+steroidal hormones, CDs+flavonoids etc.), should further calculations be performed. 

### 2.7. Møller–Plesset Perturbation Theory 2 (MP2)

Only a few (five in the period from 2014–2021) articles in which the MP2 method has been applied for CD analysis have been published. The reason is the fact that this technique is computationally more demanding than DFT and since CD complexes are relatively large, as the objects for QC studies, this method is currently not affordable for most of the computational researchers.

The most recent work in this topic was published in 2017 and concerns β-CDs with one large substituent that can either be located in or outside of the CD’s cavity [68]. The geometry optimization has been performed with the B3LYP-D3 functional but single point calculations already with both B3LYP-D3 and MP2 using various basis sets, for MP2: 6-31G* and 6-311G*. 

In another study (β-CD-sertraline) [93], MP2/6-31G(d,p) was applied for the single point calculations, although even the authors of the work state that such a small basis set does not allow one to obtain results with the required accuracy.

## 3. Preparation of Structures, Post-Processing Methods, and Some Examples 

### 3.1. Preparation of the CD Complexes for the QC Calculations

To obtain the structure of the CD complex that can be used for DFT calculations, both the structure of the chosen CD as well as the structure of the guest molecule must be prepared beforehand. The method of in silico complex preparation is also important, as it may have a major influence on the results. 

Since the crystal structures of all of the native and also some of the modified CDs can be found in the CCDC [94], they are usually used as the starting points for calculations. The structure of the guest is either simply drawn using one of the multiple available software packages or taken from the CCDC, assuming that its crystal structure has been deposited previously. 

Very important, for the accuracy of the results, is the method of preparation of the complex from its components. This must be done unless a crystal structure of the complex has been obtained or deposited previously in the CCDC, which is unfortunately quite uncommon. Usually, one of the two approaches is used. In the first one, molecular docking is applied, treating CD as a macromolecule and the guest as a ligand. In the reviewed works, the authors usually use the popular Auto Dock [95] software for that purpose. However, some other programs are also used such as Schrodinger Maestro [96] or BIOVIA Discovery Studio [97]. Surprisingly, in the reviewed studies, not much attention was being paid to the description of this part, which was justified by the fact that the initial (docked) structures would be optimized at the higher theory level. This may, however, lead to some inaccurate or even wrong results as the energetically lowest conformation obtained from the docking part may not necessarily be close to the global DFT minimum. This is nicely reflected in Figure 3, where the energetically lowest orientation obtained from molecular docking is substantially different from the experimental one, even after optimization using DFT. However, when the other pose from molecular docking was optimized using QC, much better agreement between the experimental and theoretical results was obtained. Therefore, in some studies, the authors decided to optimize not one but a few different complexes from molecular docking. While this approach is reasonable as it increases the likelihood of finding the deep minimum, it should be noticed that the time of calculations increases linearly with the number of initial structures.

Instead of molecular docking based on the molecular mechanics calculations, in some cases, the authors decided to manually dock the guest into the CD. In order to find the best pose within the cavity, the guest molecule is put in different positions along the selected axis, so that the guest has different levels of immersion into the CD’s cavity. For example, in [98], the guest was moved along the *Z*-axis from +7.5 Å to −7.5 Å with an interval of 0.3 Å. Additionally, in this particular study, the guest was rotated around the *Z*-axis by 3° from 0° to 360°. In each step, the generated systems underwent geometry optimization calculations. This type of systematic search seems to be the most accurate approach, especially for the ligands with limited conformational space, with the only drawback being the increase in the calculation time.

The other important factor that is usually neglected in the QC studies of CD complexes is the conformational flexibility of the guest molecule. The optimal conformation of the guest found *in vacuo* is not necessarily the one that it takes in the complex. To increase the likelihood of finding the deep energetical minimum, the conformational space search of the guest molecule should be performed.

Another aspect that must be taken into consideration is the host–guest molar ratio of the complex. When there are some experimental indications for a specific value, the assumption can be tested using QC calculations. Otherwise, it seems reasonable to prepare the complexes of various stoichiometry and confirm their stability via geometry optimization.

### 3.2. Description of QC Results

After the host, guest, and complex are optimized at the chosen QC level, the interaction and stabilization energies are obtained. Stabilization energy is defined as the difference between the energy of the fully optimized geometry complex and complex components: CD and guest (Equation (1)) [99]. Interaction energy is defined as the corresponding single point energy.
E_stb_ = E_cplx_opt_ − (E_CD_opt_ − E_guest_opt_)(1)

Sometimes the solvation energy is taken into account as well. It is calculated as the difference between the complex energy in water and in gas. Thermodynamic parameters (TD) are often calculated, as they give more insight into the stability of the analyzed systems.

The collected data allow us to draw conclusions, which forces determination of the complex creation: dispersion [100], van der Waals [63] interactions or hydrogen bonds [61]. TD results allow us to determine whether the complexation process is enthalpy driven [51,84,100], which is said to relate to the number and strength of the intermolecular interactions within the system, or entropy driven, which is quite rare for those complexes. Inclusion of the temperature effects allows us to observe if and how the temperature affects the complex stoichiometry, as in the case of β-CD-pentoxifilline [101].

Additionally, often IR or UV–Vis spectra or NMR chemical shifts are calculated (see Table 1) and in the majority of cases, the results are claimed to have very good agreement with the experimental data. 

Another common practice is the application of the QTAIM method (Quantum Theory of Atoms In Molecules) for the DFT-optimized complexes in order to analyze weak interactions and therefore obtain a better understanding of the complex’s structure at the molecular level. Several articles about CD complexes including this approach have been published (see Table 1).

### 3.3. Analyzed CD Complexes

Among the large variety of CDs, in the DFT studies, mainly only β-CD has been applied so far, and a variety of guests in CD complexes has been analyzed, as presented in Table 1. The guests are mainly drugs among which the antidepressants seem to be especially targeted (Table 1A.1) as well as plant derivatives with (potential) medical use (Table 1B). A separate group consists of substances that could be defined as functionalized food (Table 1C). In those cases, CDs serve to protect or even increase the antioxidative capacity of the substances in question, for example (−)-gallocatechin, (−)-catechin gallate and (−)-gallocatechin present in tea [102,103,104], or to reduce the bitter taste of coffee [105,106]. Moreover, CDs can be used as chiral selectors, and this has its reflection in the DFT articles (Table 1D). 

Some of the analyzed systems in Table 1 have been already described in the previous paragraphs as examples regarding the applied parameters and computation. Other selected examples showing a particular usefulness and applicability of the DFT methods as well as the obtained results are described below.

The examples that directly show a vast area of DFT applicability for CD systems analysis are studies of the β-CD complexes, with 8-anilinophthalene-sulfonate [84], benzyl isothiocyanthe [23], metheonine [98] or vanillina [107]. In these works, the DFT approach was used to perform geometry optimization, obtain interaction and stabilization energies, calculate thermodynamic properties, use QTAIM and NBO analysis approaches, and simulate NMR and absorption spectra. The obtained data allow the screening of possible conformations, to define the interactions (van der Waals, hydrogen bonds, etc.) determining the CD–guest interaction, rank the complexes according to their stability, complement experimental spectra, and support the signal assignment.

Further, such chemical information happens to be a crucial part of new theses. For instance, DFT calculations revealed that the nicotine forms have considerably stronger binding with β-CD rather than with Mβ-CD in the same orientation with lower complexation energy. This explains why after 21 days the remaining nicotine increased from 65.56% in pure nicotine to 89.32% and 76.22% in β-CD-nicotine and Mβ-CD-nicotine complexes, respectively [34].

Another example is imipramine and desipramine β-CD complexes [108]. DFT calculations revealed an alternative inclusion scenario: via a guest’s side chain and not via the aromatic moiety. Thus, the controversy in the experiments has been explained because such bimodal complexation increased the therapeutic effect of the substances. 

For another antidepressant, paroxetine, the DFT calculations helped to confirm the existence of a new CD inclusion polymorph: a new 2:1 stoichiometry complex has been described [109]. It is characterized by a stronger presence of the dispersion interactions and is more energetically favorable than the 1:1 complex, which improves the drug’s bioavailability. 

Again, when it comes to structural information, the DFT approach showed that in 1:2 Cu-flavonoid and 1:3 Fe-flavonoid β-CD complexes, in morin and quercetin, the 3-OH site, and in primuletin, the 5-OH site, were utilized as preferable chelation sites [25]. These data are helpful for scientists trying to obtain an effective CD-flavonoid antidiabetic formulation.

**Table 1 molecules-27-03874-t001:** Selected articles published in the years 2015–2022 on the application of DFT methods for systems that included CD. The functional and basis set information concerns CD complexes, not guests. Abbreviations used in table: DM-CD (2,6-dimethylo-CD), TM-CD (trimetylo-CD), per-M-CD (permethylated-CD), geo. opt. (geometry optimization), SP (single point calculations), NMR (1H NMR spectra simulation), NBO (Natural Bond Orbitals), BJ (Becke Johnson damping function),TD (thermodynamics calculations), n.i.p. (no information provided). In the case where the DFT application in the published research occurs only as an ONIOM component, the article has not been included in the table. The ONIOM approach along with the examples has been described in Section 2.2.

No.	CD	Guest	Functional	Basis Set	Environment	DFT Application	Ref.
**A (potential) drugs**							
1	β	(s)-2-Isopropyl-1-(o-nitrophenyl) Sulfonyl) Aziridine	B3LYP, WB97X-D, B97D3	6-31G(d)	gas, water		[110]
2	β	boron-based aromatic systems	BLYP-D3(BJ)	def2-SVP	vacuum, CPCM	geo. opt., natural bond orbital calculations (NBO), complexation energy	[100]
3	α, β, γ	alprazolam	B3LYP, M06L	def-TZVP	vacuum	geo. opt. in gas, NMR spectra	[21]
4	β	lenalidomide	B3LYP, M06-2X	6-31G(d,p)	PCM		[111]
5	β	dexamethasone	BLYP-D4	def2-TZVP	gas, water	geo. opt., complexation energy	[89]
6	β	2,2′-Bipyridine	B3LYP, wB97XD	6-31G(d)	PCM (eight solvents)	geo. opt., UV–Vis spectrum, HOMO-LUMO	[82]
7	β	2,2′-Dipyridylamine	B3LYP	6-311++G(d,p)	PCM		[112]
8		vardenafil hydrochloride	B3LYP	6-311G(2d,2p)	vacuum	geo. opt., FT-IR	[113]
9	amino-CD	doxorubicin	B3LYP	6-31G	vacuum	geo. opt., complexation energy, HOMO-LUMO, dipole moment, chemical potential, electrophilicity	[114]
10	β	5-fluorouracil	B3LYP-D3	6-31+G(d,p)	vacuum, PCM	geo. opt., complexation energy, harmonic frequency calculations	[85]
11	HP-β	2-methyl mercapto phenothiazine	B97-D3, BP86-D3	6-31G(d,p)	gas, CPCM	geo. opt., vibrational spectra, NBO, QTAIM, HOMO-LUMO	[115]
12	β	vemurafenib	ωB97XD	6-31+G(d)	vacuum, PCM	Geo. opt., vibrational spectra, MD, NBO, TD, HOMO-LUMO	[116]
13	β	procaine hydrochloride	B3LYP, M06-2X, WB97XD	6-31G(d,p)	gas, PCM	Geo. opt., NBO	[83]
14	β, SBE-β	fluorometholone, cholesterol	M06-2X	6-31G**	PCM	Geo opt., interaction energy	[117]
15	α, β, γ	chlordecone	M06-2X-D3	6-31G(d,p)	SMD	Geo. opt., QTAIM	[118]
16	β, methyl-β	nicotine	M06-2X	6-31G(d,p)	n.i.p.	Geo., opt., complexation enrgy	[34]
17	β	8-Anilinonaphthalene-1-sulfonate	B3LYP, M06-2X, WB97X-D	6-31G(d)	gas, water	Geo. opt., interaction energy, NMR, TD, NBO	[84]
18	β	benzocaine	B3LYP, CAM-B3LYP, M05-2X, M06-2X	6-31G(d,p)	PCM	Geo. opt., QTAIM, NBO, NMR, HOMO-LUMO, TD	[53]
19	β	aryl pentazole	M06-2X	6-31+G(d,p)	PCM	Geo. opt.	[119]
20	β	2,4D, dicamba pesticides	PBE1PBE (PBE0), B97-D, M06-2X	6-31G(d,p)	gas, SMD	Geo. opt.	[120]
21	Monochlorotriazinyl-β	permethrin, cyppermethrin	BLYP (geo. opt.); BLYP-D3, B3LYP-D3, M06-2X-D3 (UV–Vis)	def2-SV(P) (geo. opt.); TZVP (UV–Vis)	COSMO	Geo. opt.	[22]
22	β	dopamine	B3LYP, MPW1PW91, M05-2X, M06-2X, ωB97X-D	3-21G*	CPCM	Geo. opt., complexation energy, QTAIM, NBO	[61]
23	α	benzoate derivatives	M06L (geo. opt.); M06-2X//M06-L (SP)	6-31+G(d,p)	gas	Geo. opt.	[121]
24	α, β, γ	cholic, deoxycholic acid	B97-D, M06-2X, B3LYP	6-31G(d)	PCM	Geo. opt., interaction energy	[122]
25	α	benzoate derivatives	M06-2X//M06-L, M06-2X//BLYP, BLYP, M06-2X	6-31+G(d,p)	gas	Geo. opt., interaction energy	[123]
26	γ	cetirizine	B3LYP	def-TZVP	n.i.p.	Geo.opt., interaction energy, HOMO-LUMO, DOS, NMR	[124]
27	succinyl-β	uranium	M06-2X	6-31G(d,p)	SMD	Geo. opt.	[125]
28	β-CD, DM -β	thymidine-carbonate	B3LYP-GD2	6-31G(d,p)	PCM	Geo. opt., complexation energy, TD, HOMO-LUMO, NMR	[126]
29	β	glycyl-L-phenylalanine	B3LYP	3-21G(d)	PCM	Geo. opt., interaction energy, HOMO-LUMO	[127]
30	β	sodium salicylate	B3LYP	6-31G(d)	gas, PCM	Geo. opt., solvation energy, relative stabilization energy, complexation energy, change of volume	[128]
31	β	benzyl isothiocyanthe	B97-D3	def2-SVP	vacuum	Geo. opt., complexation energy, HOMO-LUMO, NBO, NMR	[23]
32	α	iodine solution	CAM-B3LYP	6-31*G	PCM	Geo. opt., absorption spectra, HOMO-LUMO	[129]
33	β	meta-aminophenol	M06-2X	6-31G(d,p)	IEFPCM	Geo. opt., complexation energy, HOMO-LUMO, TD, NBO	[130]
35	β	L-glutamine	B97-D3	6-31G(d)	n.i.p.	Geo. opt., complexation energy, TD, NBO, QTAIM	[131]
36	β	R and S ibuprofen	M062X	6-31G(d,p) (geo. opt.); 6-311++G(d,p) (SP)	gas, SMD	Geo. opt., solvation energy	[132]
37	α, β	thioureides	B97-D3	6–31G(d,p)		Geo. opt., interaction energy	[133]
38	β	mepivacaine	B97-D3	6-31G(d,p)	gas, SMD	Geo. opt., interaction energy, TD	[134]
39	β	L-metheonine	WB97-D3	6-31G(d)	PCM	geo. opt., interaction energy, QTAIM, TD, NMR	[98]
40	β	prazosin, losartan	B3LYP	6–311+G(d,p)	gas	Geo. opt.	[135]
41	β	olsalazine	B3LYP, WB97-D3, CAM-B3LYP (UV-vis)	6-31+G(d)	PCM	Geo. opt., ADMP	[62]
42	β	aspirin	B3LYP-D3	cc-pVDZ	gas	Geo. opt., qTAIM, NBO	[136]
43	β	quinine	B3PW91	6-311++G(d,p)	PCM	Geo. opt.	[137]
44	β	erlotinib	B3LYP	6-31+G*	n.i.p.	Geo. opt., harmonic frequencies, HOMO-LUMO	[138]
45	γ	rocuronium, vecuronium	B3LYP	6–31+G(d,p)	n.i.p.	Geo. opt., NBO, HOMO-LUMO	[139]
46	α, β, γ	cathinone	M05-2X	6-31G(d)	gas, CPCM (water, chloroform, methanol)	Geo. opt., QTAIM, NBO, IR spectra, TD	[140]
47	α	CO2	B3LYP	G-31G*	PCM	NMR	[141]
48	β	flutafemic acid	B3LYP, M05-2X	6-31G(d)	vacuum, water	Geo. opt., complexation energy, TD, NMR	[142]
49	2-HP-β	Cu (II) and Fe (III) complexes of quercetin, morin, primuletin	B3LYP	6-311++G**	n.i.p.	Geo. opt., complexation energy, HOMO-LUMO	[25]
50	β	6-thioguanine, 6-mercaptopurine	B3LYP	6-31+g(d,p)	IEFPCM (DMSO)	Geo. opt., interaction energy, TD	[37]
51	β	N-(2-chloroethyl),N -nitroso,N′,N′-dicyclohexylsulfamid	B3LYP	6-31G(d)	PCM (DMSO)	Geo. opt., NBO, QTAIM	[143]
52	β	benzaldehyde	B97-D	6-31G(d,p) (geo. opt.); 6-311++G(2d,p) (SP)	gas, SMD	Geo. opt., interaction energy, TD	[144]
53	α	chitibiose	M06-2X	6-311++G**	n.i.p.	Geo. opt., NBO, QTAIM	[145]
54	α	hydrated and nonhydrated IIA/IIB group metal cations	M06-2X	6-31G(d,p)	gas, PCM	Geo. opt., interaction energy, TD	[146]
55	β	nabumetone	WB97X-D, B97-D, B3LYP, M05-2X, M06-2X	6-31G(d)	IEFPCM	Geo. opt., NBO, QTAIM	[40]
56	β	propranolol	B3LYP, ωB97XB (ONIOM)	6-31+G(d)	gas, IEFPCM, explicit solvent effect: explicit water molecules inside of the complex	Geo. opt., interaction energy, ADMP, TD	[59]
57	functionalized CDs	8-hydroxyquinoline ligands	B3LYP	6-31G**	n.i.p.	Geo. opt.	[147]
58	β	pentoxifilline	M06-2X	6-31g(d,p)	gas	Geo. opt., NBO, HOMO-LUMO	[101]
59	β	p-nitropenthyl acetate	B3LYP	6-31G(d,p)	n.i.p.	Geo. opt., interaction energy, NBO, HOMO-LUMO	[148]
60	β	norfloxacin	B97D (geo. opt.), B3LYP (SP, NMR)	6-31G(d,p)	IEFPCM	Geo. opt., interaction and stabilization energy, NMR, TD	[149]
**A1. Antidepressants**							
61	β	paroxetine	B3LYP (geo. opt.); B97D (SP)	6-31+G* for H, N, O and 4-31G for C	vacuum	Geo. opt., interaction energy, TD	[109]
62	2,6-DM-β	mianserin	B3LYP-GD2 (geo. opt.); M05-GD3, M06-GD3, M062X-GD3, ωB97XD, mPW1PW91, M11 (SP)	6-31G(d,p)	PCM, vacuum	Geo. opt., interaction energy, NMR	[150]
63	β	sertraline HCl, fluoxetine HCl	B3LYP	6-31+G* for H, N, O and 4-31G for C	gas	Geo. opt., interaction energy	[151]
64	β	protriptyline, maprotiline	B3LYP	6-31+G* for H, N, O and 4-31G for C	vacuum	Geo. opt., interaction and stabilization energy	[152]
65	β	clomipramine, doxepin	B3LYP	31+G(d) for H, N, O, Cl, and 4-31G for C	gas	Geo. opt., interaction energy	[153]
66	β	desipramine, imipramine	B3LYP	6-31þG(d) for H, N, O and 4- 31G for C	gas, implicit solvent (water)	Geo. opt., interaction and stabilization energy	[108]
67	β	amitryptyline, nortryptiline	n.i.p.	6-31+G* for H, N, O, Cl and 4-31G for C	vacuum, SMD	Geo. opt., interaction and stabilization energy	[154]
**B. Plant derivatives**							
68	HP-β	thymoquinone	B3LYP-D2, B3LYP-D3	6-31G(d,p)	PCM	Geo. opt., NBO, QTAIM, HOMO-LUMO, NMR	[155]
69	α	β-carotene	B3LYP	cc-pVDZ	vacuum	Geo. opt., interaction energy, Raman spectra	[156]
70	γ	3-hydroxyflavone	PBE0	def2-SV	PCM	Geo. opt., HOMO-LUMO, IT spectra	[157]
71	β	vanillina	B3LYP, ωB97xD, M06- 2X	6-311G(d,p)	vacuum, CPCM	Geo. opt., interaction energy, NMR, HOMO-LUMO, NBO, QTAIM, UV–Vis	[107]
72	β	alfa-terpineol	B3LYP (for UV–Vis), B3LYP/CAM, M062X, WB97-D3	6-311G(d,p)	vacuum, CPCM	Geo. opt., complexation energy, NBO, QTAIM, TD, UV-vis	[158]
73	TM-β, β	naringenin	B3LYP, M06-2X, wB97X-D	6-31G(d)	vacuum	Geo. opt., interaction energy, NBO, QTAIM, NMR, HOMO-LUMO	[159]
74	2,6-DMβ, 2HP-β, 2,6-DH-β, β	eucalyptol	M06-2X	6-31G(d,p)		Geo. opt., interaction energy	[160]
75	β	fisetin	M06-2X	6-31G(d,p)	gas, PCM	Geo. opt., interaction energy	[161]
76	β	gallic acid	B97-D3	6-31G*, for GIAO: 6-311++g**	gas, solvent	Geo. opt., HOMO-LUMO, NBO, NMR	[162]
77	β	gabapentin	B3LYP-D3	6- 31G(d)	vacuum, PCM	Geo. opt., interaction energy, NBO, HOMO-LUMO	[163]
78	Β, γ	tropane alkaloids	B3LYP	6-31+G(d,p)	PCM	Geo. opt., interaction energy, NMR	[164]
79	β	coumarins	EDF2	6-311G(d,p)	PCM	Geo. opt.	[165]
80	2-HP-β	quercetin	B3LYP	6-31G*		Geo. opt.	[166]
81	β	carvacrol, thymol	B3LYP	6-31G, 6-31+G(d)	SMD	Geo. opt., interaction energy, NBO, HOMO-LUMO	[167]
82	β	thymol	B3LYP, PBEPBE, CAM-B3LYP	6-31G(d,p)	PCM	Geo. opt., interaction energy, UV–Vis	[168]
83	β	carvacrol	B3LYP, M05-2X	6-31G(d)	PCM	Geo. opt., HOMO-LUMO, NBO	[42]
**C. Functionalized food**							
84	β	(−)-gallocatechin, (−)-catechin gallate, (−)-gallocatechin gallate	B3LYP	6-31+G* for H, O and 4-31G for C	gas	Geo. opt., interaction and stabilization energy	[102,103]
85	β	(−)-epigallocatechin, (−)-epigallocatechin gallate	B3PW91	cc-pVDZ	gas	Geo. opt., interaction energy	[104]
86	β	catechol derivatives: protocatechuic aldehyde, protocatechuic acid	B3LYP	6-31+G* for H, O and 4-31G for C	gas (geo. opt.), implicit solvent (TD)	Geo. opt., interaction energy, TD	[169]
87	β	oleuropein, hydroxytyrosol, tyrosol	n.i.p.	6-31+G* for H, O and 4-31G for C	gas	Geo. opt., interaction energy, TD	[170]
88	β	chlorogenic, caffeic, quinic acids	B3LYP	6-31+G* for H, O and 4-31G for C	gas	Geo. opt., interaction energy, TD	[105,106]
**D. CD as a chiral selector**							
89	β	D- and L-penicillamine	B3LYP-D3 (geo. opt.); M062X-D3, xB97X-D, B3LYP-D3 (interaction energy)	6-31G(d, p) (geo. opt.); G-311+G(d,p) (interaction energy)	water	Geo. opt., interaction energy	[171]
90	metal-ion coupled β	D- and L-penicillamine	DFT, M062X	6-31G(d,p)	vacuum	Geo. opt.	[172]
91	β	R- and S-propranolol	B3LYP	6-311+G(d,p)	vacuum	Geo. opt., vibrational spectra	[173]
92	per-M β	D- and L-isoleucine	B3LYP (geo. opt.), wB97X-D (IR)	6-31G*, 6-311G**	gas	Geo. opt., interaction energy, IR spectra, TD	[174]
93	per-M β	D- and L-alanine	B3LYP, wB97X-D, M06-2X	6-31G**, 6-311G**		Geo. opt., IR spectra	[175]
94	2,3,6-TM-β	cis-(2S,4R) and -(2R,4S) ketoconazole	B3LYP	6-311G(d,p)	gas (geo. opt.), PCM (SP)	Geo. opt., interaction energy	[36]
95	2-HP-β	abacavir enentiomers	PBE	6-31G*	PCM	Geo. opt., interaction energy	[176]

## 4. Conclusions

The number of studies concerning CDs complexes in which the theoretical calculations at the QC level have been used has constantly increased since the beginning of the 21st century. Solely for the DFT-based works in this topic, the number of published articles has already exceeded 300. While this number is still relatively low, when compared to the amount of reported molecular dynamics simulations at the molecular mechanics level [12], the reviewed works reveal that the application of QC calculations in the studies of CD complexes can be essential, providing the results unobtainable by any other method, both experimental and computational.

Initially, for those kind of studies, less computationally demanding semi-empirical methods have been applied (mostly PM3, PM6, and PM7). However, since 2015, each year there have been more papers in which DFT has been chosen instead of the semi-empirical approach. Nevertheless, even in some works published after 2020, the authors have found that the PM6 or PM7, when used with appropriate dispersion correction, can provide results with similar accuracy to those obtained using DFT.

Regarding calculations of geometries and interaction energies with DFT methods, in most of the reviewed works, the inclusion of dispersion correction was found to be crucial to obtain accurate energies, irrespective of the basis set and functional used. As for the functionals, there is no surprise that B3LYP, which is the most commonly applied one in the field of organic molecule calculations, is also the one that is most extensively used for the studies of CD complexes. However, in some works where the authors have used wB97X or M06-2X instead, different results have been obtained both in terms of predicted geometry as well as stability ranking.

When preparing the complexes, some authors prefer to manually dock the molecules, systematically moving the guest towards the CD cavity and rotating it; however, in most of the works, the molecular mechanics docking procedure has been applied, usually employing the popular and freeware Auto Dock. It should be noticed that the best pose from docking was not always the one with the lowest DFT energy; therefore, at least a few different poses should be optimized in order to achieve credible results.

Though in most of the reviewed works the authors have limited their calculations solely to geometry optimization of one or a few conformations, in some of the articles, the complex properties have been computed. Successful application of DFT methods include prediction of UV–Vis, IR, and NMR spectra as well as HOMO-LUMO and NBO calculations.

The major problem with the DFT calculations seems to be the solvent treatment. In the vast majority of the reviewed works, the authors decided to apply an implicit solvation model, usually PCM or SMD. However, other studies have shown that the role of water in the complex formation can be crucial as the water-mediated hydrogen bonds between the host and guest have been observed many times. Further, the role of water release from the cyclodextrin cavity during the complexation significantly affects the thermodynamics of such a process, which can be modeled accurately only using explicit water models.

Finally, it should be emphasized that even when carefully choosing the appropriate DFT method (applied functional, basis set, solvation scheme, dispersion correction, credible initial conformation), the obtained results can still be different from the corresponding experimental ones. This is due to the high flexibility and dynamics of most of CD complexes. Therefore, it seems reasonable to explore the application of ab initio molecular dynamics simulations. While, at this moment, it may be computationally not affordable for many researchers, combining the benefits of molecular dynamics simulations with the accuracy of DFT calculations seems to be the solution to obtain even more accurate results.

## Figures and Tables

**Figure 1 molecules-27-03874-f001:**
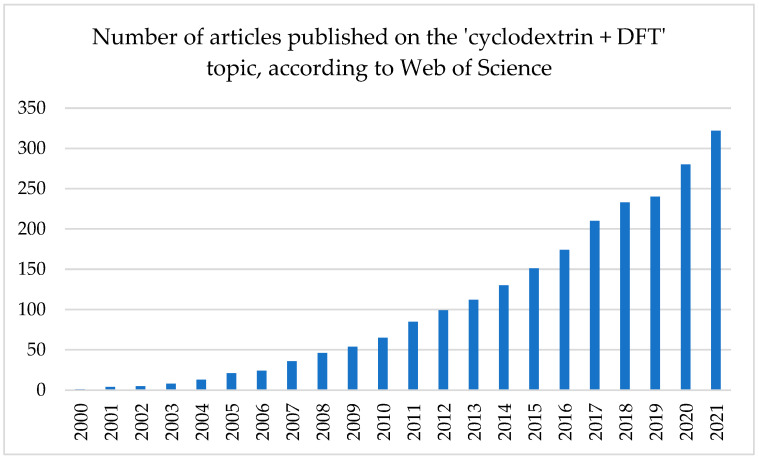
Number of the search results for the ‘cyclodextrin AND DFT’ phrase in the Web of Science. Each column shows the number of articles in the given year and all years before. For example, the column entitled ‘2010’ depicts the number of articles published in the period 2000–2010, including 2010.

**Figure 2 molecules-27-03874-f002:**
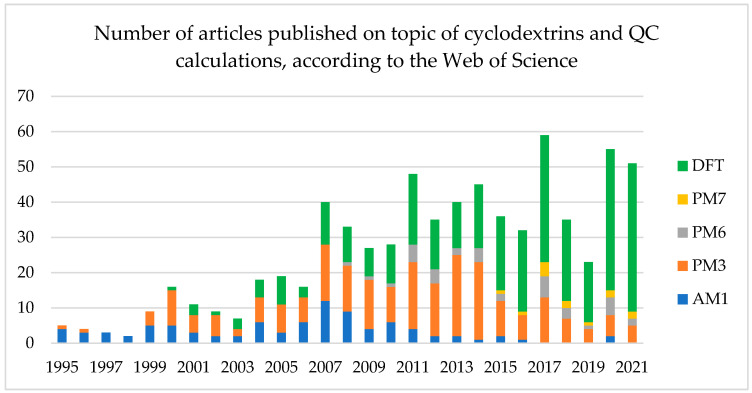
Changes in the number of articles published on the topic of cyclodextrin and either different semi-empirical or DFT methods over the years.

**Figure 3 molecules-27-03874-f003:**
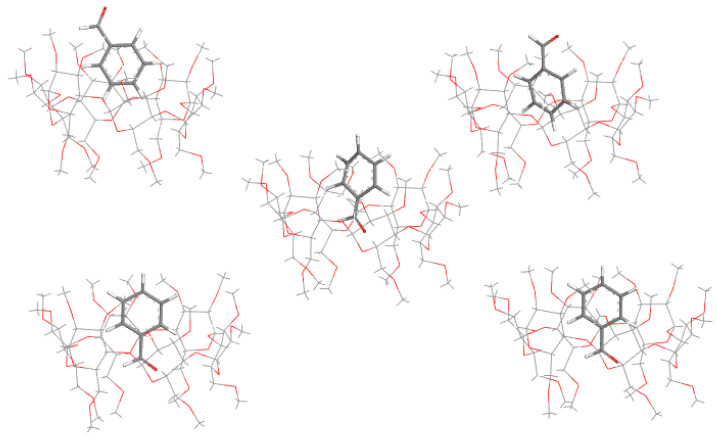
Comparison between the structures of the αCD complex with benzaldehyde. Top left: the best pose from molecular docking; top right: the best pose from molecular docking after optimization using DFT; bottom left: one of the poses obtained from molecular docking; bottom right: “bottom left” structure after optimization using DFT; middle one: experimental structure (CSDC ref. code: BOHWUQ). It should be noted that while the top left structure has energy lower than the bottom left by 3.4 kcal/mol, the top right structure has energy higher than the bottom right by 4.2 kcal/mol. Source: author’s archive.

## Data Availability

Not applicable.

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
