# Peer review of "Current Status of Quantum Chemical Studies of Cyclodextrin Host–Guest Complexes"

_molecules, 2022, doi:10.3390/molecules27123874_

Round 1
Reviewer 1 Report
This manuscript projects an overview of various quantum chemical methods application such as semi empirical, DFT, MP2 in the studies of cyclodextrin host-guest complexes. This review will be of immense importance not only to the quantum chemists engaged in this area but also to the general readers as it covers all. The review has been presented in a perfect and concise/updated manner. I recommend acceptance of this manuscript in Molecules. I suggest authors to change title with better, include molecular structures of CDs as well as their host-guest phenomenon with example as a figure and add table of contents image (graphical abstract).
Author Response
Comment:
This manuscript projects an overview of various quantum chemical methods application such as semi empirical, DFT, MP2 in the studies of cyclodextrin host-guest complexes. This review will be of immense importance not only to the quantum chemists engaged in this area but also to the general readers as it covers all. The review has been presented in a perfect and concise/updated manner. I recommend acceptance of this manuscript in Molecules. I suggest authors to change title with better, include molecular structures of CDs as well as their host-guest phenomenon with example as a figure and add table of contents image (graphical abstract).
Response:
Thank you very much for the nice comments and the significant effort needed to create this review. We’ve found all of your suggestions very helpful in improving the quality of our manuscript. The title has been changed to “Current status of quantum chemical studies of cyclodextrin host-guest complexes” as suggested by the Reviewer #4. Besides, graphical abstract has been prepared and submitted. Enclosed, please find the corrected version of the article.

Reviewer 2 Report
This manuscript reviews the application of computational methods in the study of cyclodextrin host-guest complexes. It is well written and discussed and I recommend acceptance in its current form.
Author Response
Comment:
This manuscript reviews the application of computational methods in the study of cyclodextrin host-guest complexes. It is well written and discussed and I recommend acceptance in its current form.
Response:
Thank you very much for a nice comment. Enclosed, please find the corrected version of the article.

Reviewer 3 Report
This manuscript reviewed the application of quantum chemistry methods (DFT, SE and MP2) in the research of cyclodextrin host-guest complexes. This is a thoroughgoing review report. It not only provided the information about method choice but also the theoretical details about the QC methods with a long history. It could help researchers select the proper QC method to research cyclodextrin host-guest complexes. In my opinion, this manuscript can be published on molecules after several improvement. I hope the improvements will help the readers. My suggestions are listed below.
1) Could please mention DFTB and force field in your manuscript? DFTB is more popular than SE methods (AM1, PM6, PM7, …). Force field is also a valuable method. Just add few sentences is acceptable.
(https://pubs.acs.org/doi/full/10.1021/acs.jctc.7b00359) Force field.
2) If I want to research cyclodextrin host-guest complexes, I will read benchmark articles in advance. Could you please search more benchmark articles about cyclodextrin host-guest complexes?
I found one, https://pubs.acs.org/doi/full/10.1021/acs.jctc.5b00296
Author Response
Comment:
This manuscript reviewed the application of quantum chemistry methods (DFT, SE and MP2) in the research of cyclodextrin host-guest complexes. This is a thoroughgoing review report. It not only provided the information about method choice but also the theoretical details about the QC methods with a long history. It could help researchers select the proper QC method to research cyclodextrin host-guest complexes. In my opinion, this manuscript can be published on molecules after several improvement. I hope the improvements will help the readers. My suggestions are listed below.
1) Could please mention DFTB and force field in your manuscript? DFTB is more popular than SE methods (AM1, PM6, PM7, …). Force field is also a valuable method. Just add few sentences is acceptable.
(https://pubs.acs.org/doi/full/10.1021/acs.jctc.7b00359) Force field.
2) If I want to research cyclodextrin host-guest complexes, I will read benchmark articles in advance. Could you please search more benchmark articles about cyclodextrin host-guest complexes?
I found one, https://pubs.acs.org/doi/full/10.1021/acs.jctc.5b00296
Response:
Thank you very much for a nice comment and the significant effort needed to create this review. We’ve found all of your suggestions very helpful in improving the quality of our manuscript. Enclosed, please find the corrected version of the article.
- We have added a paragraph about the application of DFTB methods in the analysis of cyclodextrin host-guest complexes.
An approach which can be positioned between semi-empirical methods and DFT is Density Functionals Tight Binding Self Consistent Charge method (DFTB-SCC, shortly referred here to as DFTB) which is often described as DFT approximation [DOI:10.1007/s10847-005-9030-9]. According to the Web of Science just a few articles referring to DFTB and CD complexes have been published. However those works show a relatively wide spectrum of possible DFTB applications. In the oldest works [DOI:10.1007/s10847-005-9030-9; DOI:10.1016/j.peptides.2007.08.011] DFTB is applied to verify the experimental NMR results and deliver some additional structural information. In the first case [DOI:10.1007/s10847-005-9030-9], method’s application confirms which conformation of spironolactone is preferred in the complex. In the second case [DOI:10.1016/j.peptides.2007.08.011], DFTB confirms that in the analyzed peptide the tyrosine is a favoured residue to access the CD’s cavity. In this second study, DFTB is applied within the QM/MM approach. In both of the described cases authors claim good agreement of the obtained computation data with the experimental ones.
More complex is situation described in [DOI:10.1016/j.tet.2017.07.030], where the topic is self-inclusion (in own cavity) of the CD’s substituents. Here, DFTB has been compared with DFT approach. DFTB is said to be method of choice as it predicts the stability order of the analyzed complexes properly, delivers the energy data which are close to the DFT ones and is a faster than DFT. Both the option including dispersion correction and the option without this correction have been tested in DFTB and in DFT. The results plainly show that in both cases application of the correction is necessary. In case of DFTB, the dispersion corrected version of calculations end up with lower complexation energies and the order is maintained. Nevertheless, the authors emphasize that the application of an empirical dispersion correction may significantly overestimate dispersion interactions and therefore a comparison with a rigorous DFT method should be done.
Some drawbacks of DFTB have been pointed out in the work which targeted the largest CD for which the crystallographic structure is known [DOI:10.1021/jp208927v]. There the heavy atoms RMSD between the optimized structure and the crystallographic one were between 0.89 Å and 1.35 Å for DFT, depending on the parameters applied, with the best results for B3LYP functional and 0.95 Å for DFTB. However, even if the overall RMSD looks good, large discrepancies in the angles sizes when compared with the experiment have been reported in case of DFTB approach as opposed to all DFT methods.
In turn, the article from 2018 utilizes DFTB approach to analyze the tautomerization process during encapsulation of genistein in CD [DOI:10.1016/j.molliq.2018.05.109]. The results are clear and deliver important information on the complexation, namely: ‘DFTB-based MD simulations reveal that spontaneous keto-enol tautomerization occurs even within a hundred picoseconds, which suggests that the encapsulated genistein is complexed in the ordinary enol form of the drug molecule.’.
- We totally agree with the Reviewer. Unfortunately, as those kind of calculations are computationally demanding only a few benchmark studies have been published as far. We have added them to the review with the suitable comment.
(…) Taking into account all what has been written above, in order to make a fully jus-tified selection of the parameters for the DFT calculations, a cross-study including dif-ferent but most used functionals, dispersion corrections (presence or absence), basis sets and environment (gas or solvent, type of solvation) should be performed. To the authors best knowledge, so far no such study has been ever undertaken on any CD-including system. The already published benchmark studies are quite uncommon and usually focus on modification of one of the parameters, either functional, dispersion correction or solvation method [10.1016/j.tet.2017.07.030 ; 10.1021/acsomega.0c01059 ; 10.1080/10610278.2017.1401074 ; 10.1021/acs.jctc.5b00296 ].

Reviewer 4 Report
The review by Mazurek and Szeleszczuk aims to consider the application of QC methods in the studies of cyclodextrin complexes. Authors processed a huge literature material that resulted in the conclusions on the most frequently and the most accurate QC approximations used to analyze complexes of CDs. I think that the review can be published as it would be of a great importance for the CD community and, possibly, for other scientists dealing with host-quest interactions and non-covalent interactions in general. However, some points should be addressed at the current stage.
First of all, I would like to correct authors and remind them that the DFT methods are the functionals (strictly speaking, dispersion corrections are also functionals) and not the combination of functionals / basis sets / solvent model (SMD, PCM, ONIOM etc.) / ensemble description (MD, Monte-Carlo, harmonic app. etc.). I think this point is the main problem of the review in its current form. The corresponding reorganization of the review would clearly facilitate its perception for a common reader. The specific suggestions and other minor questions are listed below:
1. Page 5, lines 159-162. This should be rewritten as the better description of weak interactions by PM7 causes the decrease of the errors in the prediction of properties. Not the other way around.
2. Page 5, lines 169-176. It would be better to add the comparison of ADMP with molecular mechanics MD. I understand that this can be hardly done for the available literature on CDs but a few words to describe the ADMP method would be useful, especially accounting for the review on MM MD studies of CDs published earlier.
3. All in all, the value of MD techniques is not disclosed by authors. I suggest to pay some attention (may be in the introduction) to the general roles of different PES scanning techniques and different types of property calculations. The MD is worthy to be separated from the semi-empirical section (may be after all the methods, solvent models and basis sets will be discussed) as even the APMD is not as semi-empirical as PM6 and, strictly speaking, the MD is the type of calculation not the method of calculation.
4. Page 5, lines 195-196. Electron density functional define not only the XC energy but the whole electronic energy. XC functionals do define the XC energy term. Finally, despite the main problem of DFT is the choice of a proper XC functional, the functionals approximating the kinetic energy also exist.
5. Page 5, line 206: B97D3 instead of 97D3? At the cited paper there is no information on this combination.
6. Page 7. The use of ONIOM and other QM:QM and QM:MM combinations to model solvation effects are worthy to discuss here in the corresponding ‘solvation’ section rather than in the 2.2.2. paragraph. It would be better to separate the ‘solvent’ and ‘basis sets’ sections from the DFT paragraph as the proper choice of both is necessary no matter what theory (density functionals or wavefunctions or density matrices etc.) is used.
7. I think the 2.2.2. paragraph is better to be placed in the beginning of the review, where the objects of modelling (at least given as a structure scheme) and the types of calculations (see point 3) would be also described. Also, the role of periodic calculations for CDs modelling could be elaborated to a greater extent.
8. I don’t fully understand why the calculation of stabilization energy, forces, TD parameters and QTAIM are discussed in the section entitled as “Preparation of the CD-complexes for the DFT calculations.” This may be better to be separated into the “preparation of the CD-complexes for the QC calculations” and “description of QC results” or something like that.
9. I suggest to rename the chapter 3 (“Analyzed systems and reasons for the DFT application”). I understand that the DFT is a good choice but there are plenty of lightweight post-HF methods (like DLPNO-CCSD, DMRG etc.) which will hopefully be more available in the nearest future. I also don’t understand what the ‘reasons for the DFT application’ means. Might be, the “Preparation of structures, post-processing methods and some examples” would describe better the content of this chapter.
10. The Table must be clearly formatted. It would be better to place it between the chapters.
Finally, I think that the title of the review does not match its content to the fullest. Authors pay a lot of attention for the QC approximations while the applications of QC to the analysis of CDs complexes are considered to a lesser extent (at least, even the organization of the review corresponds to the QC approximations and not to the applications). It is just a suggestion but may be the “Current status of quantum chemical studies of cyclodextrin host-guest complexes” performs better.
Author Response
Comment:
The review by Mazurek and Szeleszczuk aims to consider the application of QC methods in the studies of cyclodextrin complexes. Authors processed a huge literature material that resulted in the conclusions on the most frequently and the most accurate QC approximations used to analyze complexes of CDs. I think that the review can be published as it would be of a great importance for the CD community and, possibly, for other scientists dealing with host-quest interactions and non-covalent interactions in general. However, some points should be addressed at the current stage.
First of all, I would like to correct authors and remind them that the DFT methods are the functionals (strictly speaking, dispersion corrections are also functionals) and not the combination of functionals / basis sets / solvent model (SMD, PCM, ONIOM etc.) / ensemble description (MD, Monte-Carlo, harmonic app. etc.). I think this point is the main problem of the review in its current form. The corresponding reorganization of the review would clearly facilitate its perception for a common reader. The specific suggestions and other minor questions are listed below:
Response:
Thank you very much for the significant effort needed to create this review. We’ve found all of your suggestions very helpful in improving the quality of our manuscript. We entirely agree with the Reviewer that the DFT methods are, indeed, functionals. However, during the choice of the computational method one has to make some decisions, like the choice of functional, dispersion correction, basis set, solvation scheme etc. Therefore, we wanted to arrange the review in a way that would facilitate making those decisions. Below, please find the direct responses to your comments.
Comment:
- Page 5, lines 159-162. This should be rewritten as the better description of weak interactions by PM7 causes the decrease of the errors in the prediction of properties. Not the other way around.
Response:
We fully agree, this part has been rewritten. “In contrast to PM6, in PM7 description of dispersion interactions and hydrogen bonding has been improved, and consequently the errors associated with modelling large molecules an complexes have been reduced [35]. Besides, the description of properties like heat of formation or height of reaction barrier has been improved [45].”
Comment:
- Page 5, lines 169-176. It would be better to add the comparison of ADMP with molecular mechanics MD. I understand that this can be hardly done for the available literature on CDs but a few words to describe the ADMP method would be useful, especially accounting for the review on MM MD studies of CDs published earlier.
Response:
The Reviewer is correct, to the best of our knowledge there are no published studies comparing the results of MM MD and ADMP on the cyclodextrin hos-guest complexes. As suggested, we have added more information about the ADMP with a suitable reference.
“The ADMP method has a number of attractive features. Systems can be simulated by accurately treating all electrons or by using pseudopotentials. Through the use of a tensorial fictitious mass and smaller values of the mass, reasonably large time steps can be employed, and lighter atoms such as hydrogens need not be replaced with heavier isotopes. A wide variety of exchange-correlation functionals can be utilized, including hybrid density functionals. [10.1021/jp034633m]“
Comment:
- All in all, the value of MD techniques is not disclosed by authors. I suggest to pay some attention (may be in the introduction) to the general roles of different PES scanning techniques and different types of property calculations. The MD is worthy to be separated from the semi-empirical section (may be after all the methods, solvent models and basis sets will be discussed) as even the APMD is not as semi-empirical as PM6 and, strictly speaking, the MD is the type of calculation not the method of calculation.
Response:
The part about the MD simulations has been added. However, we didn’t want to extend this section for two reasons. First of all there are only a few studies of CD host-guest complexes using aiMD. Second, we have recently reviewed the MM MD studies [10.3390/ijms22179422] and a reference to this work can be found in the present article.
Comment:
- Page 5, lines 195-196. Electron density functional define not only the XC energy but the whole electronic energy. XC functionals do define the XC energy term. Finally, despite the main problem of DFT is the choice of a proper XC functional, the functionals approximating the kinetic energy also exist.
Response:
We agree with the Reviewer, this has been corrected.
Comment:
- Page 5, line 206: B97D3 instead of 97D3? At the cited paper there is no information on this combination.
Response:
Thank you for pointing this, it has been corrected. In the cited paper the author has used both the long-range corrected, range-separated functional with D2 empirical dispersion correction wB97XD, as well as a pure functional, B97, with the Grimme's D3BJ dispersion [93], B97D3.
Comment:
- Page 7. The use of ONIOM and other QM:QM and QM:MM combinations to model solvation effects are worthy to discuss here in the corresponding ‘solvation’ section rather than in the 2.2.2. paragraph. It would be better to separate the ‘solvent’ and ‘basis sets’ sections from the DFT paragraph as the proper choice of both is necessary no matter what theory (density functionals or wavefunctions or density matrices etc.) is used.
Response:
The “ONIOM” part has been moved to the separate paragraph (2.2.), as instructed in the next comment. The “solvation” section has been converted into separate paragraph (2.5.).
Comment:
- I think the 2.2.2. paragraph is better to be placed in the beginning of the review, where the objects of modelling (at least given as a structure scheme) and the types of calculations (see point 3) would be also described. Also, the role of periodic calculations for CDs modelling could be elaborated to a greater extent.
Response:
The previously named “2.2.2.” paragraph has been moved at the beginning of the review, just below Figure 2. We agree that it fits better there. Besides, we have added some information on the periodic calculations of CD complexes.
“Besides, it should be noticed that to increase the accuracy of DFT calculations in solid state, periodic boundary conditions are often applied, using crystal unit cells as simulation boxes. During the computations only the properties of the original unit cell need to be calculated and then propagated in the chosen dimensions. However, to per-form such computations the crystal structure of the studied object is mandatory. More information on such calculations can be found in a recent review [10.3390/pharmaceutics12050415] with an example of such calculations for CDs presented here: [10.1107/S2052520614022902].”
Comment:
- I don’t fully understand why the calculation of stabilization energy, forces, TD parameters and QTAIM are discussed in the section entitled as “Preparation of the CD-complexes for the DFT calculations.” This may be better to be separated into the “preparation of the CD-complexes for the QC calculations” and “description of QC results” or something like that.
Response:
This has been corrected, following directly the Reviewer’s instructions. This section has been divided into two parts: “Preparation of the CD-complexes for the QC calculations” and “Description of QC results”.
Comment:
- I suggest to rename the chapter 3 (“Analyzed systems and reasons for the DFT application”). I understand that the DFT is a good choice but there are plenty of lightweight post-HF methods (like DLPNO-CCSD, DMRG etc.) which will hopefully be more available in the nearest future. I also don’t understand what the ‘reasons for the DFT application’ means. Might be, the “Preparation of structures, post-processing methods and some examples” would describe better the content of this chapter.
Response:
Thank you for this advice. As suggested, the title of the chapter has been changed to “Preparation of structures, post-processing methods and some examples”.
Comment:
- The Table must be clearly formatted. It would be better to place it between the chapters.
Response:
The table was moved and placed between the chapters 3 and 4.
Comment:
Finally, I think that the title of the review does not match its content to the fullest. Authors pay a lot of attention for the QC approximations while the applications of QC to the analysis of CDs complexes are considered to a lesser extent (at least, even the organization of the review corresponds to the QC approximations and not to the applications). It is just a suggestion but may be the “Current status of quantum chemical studies of cyclodextrin host-guest complexes” performs better.
Response:
Thank you for this suggestion. As instructed, the title has been changed to “Current status of quantum chemical studies of cyclodextrin host-guest complexes”.

Round 2
Reviewer 4 Report
Authors do their best to address all points indicated by this reviewer. I think now the review is even more worthy to publish. I wish the authors every success in the future.
My recommendation is 'publish as is'